

# Targeting multi-loop integrals with neural networks

Ramon Winterhalder[1,2,3], Vitaly Magerya[4], Emilio Villa[4], Stephen P. Jones[5],
Matthias Kerner[4,6], Anja Butter[1,2], Gudrun Heinrich[2,4] and Tilman Plehn[1,2]

**1** Institut für Theoretische Physik, Universität Heidelberg, Germany
**2** HEiKA - Heidelberg Karlsruhe Strategic Partnership, Heidelberg University,
Karlsruhe Institute of Technology (KIT), Germany
**3** Centre for Cosmology, Particle Physics and Phenomenology (CP3),
Université catholique de Louvain, Belgium
**4** Institut für Theoretische Physik, Karlsruher Institut für Technologie, Germany
**5** Institute for Particle Physics Phenomenology, Durham University, UK
**6** Institut für Astroteilchenphysik, Karlsruher Institut für Technologie, Germany

## Abstract

Numerical evaluations of Feynman integrals often proceed via a deformation of the integration contour into the complex plane. While valid contours are easy to construct, the numerical precision for a multi-loop integral can depend critically on the chosen contour. We present methods to optimize this contour using a combination of optimized, global complex shifts and a normalizing flow. They can lead to a significant gain in precision.

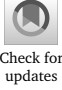

## Content

# 1  Introduction

High-precision predictions based on quantum field theory are the cornerstone in the LHC research program probing fundamental physics and the way towards identifying physics beyond the Standard Model. Improved analysis techniques, better controlled systematics and the planned 25-fold luminosity increase during the LHC Runs 3 and 4 translate into a major challenge for the corresponding theory calculations and simulations [1]. Furthermore, the prospect of a future lepton collider will require the calculation of electroweak corrections with many different mass scales, where the underlying multi-loop integrals are out of reach for analytical approaches so far, while numerical methods are promising [2].

For all aspects of LHC simulations, including numerical approaches to loop integrals and amplitudes, computing time and algorithmic efficiency are essential. Therefore, it is crucial that we investigate new methods which have the potential to improve the numerical efficiency, in particular machine learning (ML) methods. Along the established LHC simulation chain, machine learning has already shown great promise when it comes to faster and more precise predictions [3]. This includes phase space integration [4, 5], phase space sampling [6–9], amplitude evaluation [10–13], event subtraction [14], event unweighting [15, 16], parton showering [17–20], parton densities [21, 22] or particle flow descriptions [23, 24]. Full neural network-based event generators [25–30] can be used to invert the simulation chain and unfold detector effects as well as QCD jet radiation [31–33]. An important issue in applying machine learning to LHC predictions is uncertainty control and quantification, which is being addressed with increasing success [9, 11, 30, 34–36].

Essentially all of these ML-applications are driven by three properties of neural networks: they are very flexible in what they describe and how they are trained, they provide an excellent interpolation, and they are extremely fast once trained. These numerical advantages naturally lead us to investigate where ML could be useful in multi-loop calculations.

We present a first application of modern neural networks in loop integrals. Our starting points are Feynman integrals in a parametric representation, where the loop momenta have been integrated out analytically by the standard procedure, leading to the two Symanzik polynomials $\mathcal{U}$ and $\mathcal{F}$, see e.g. [1] for a description. Such integrals often have poles which manifest themselves as powers of $1/\epsilon$ in dimensional regularization and can be factorized efficiently with sector decomposition [37, 38]. After factorizing the poles, integrable singularities, related for example to thresholds, remain. They can be dealt with by a deformation of the integration contour into the complex plane [39–42]. An automated procedure to do so has been implemented for the first time in SECDEC [43] and has been refined in SECDEC-3 [44] and pySECDEC [45–47].

The deformation of the integration contour can be performed in many ways, the only requirement is that no pole is crossed by the deformation. Applications to multi-loop integrals with a certain complexity show that the numerical precision can vary by orders of magnitude depending on the choice of a particular contour. In this work we present methods to optimize the choice of the contour based on neural networks.

In Section 2 we briefly review the construction of the Feynman parametric representation of multi-loop integrals, which forms the starting point of our investigations, as well as the contour deformation procedure employed in pySECDEC. In Section 3 we describe our new approach to contour deformation based on neural networks and show results for several examples, before we give an Outlook.

## 2 Multi-loop Feynman integrals

For our study of ML-methods we focus on the numerical evaluation of integrals in the Feynman-parameter representation. Before we show how neural networks can improve the numerical evaluation of such integrals, we briefly review their definition and the way they are evaluated in pySECDEC.

**Feynman parametrization**

A generic scalar Feynman integral in $D$ space-time dimensions with $L$ loops and $N$ propagators of arbitrary powers $\nu_j$ can be represented by

$$G = \int_{-\infty}^{\infty} \left( \prod_{l=1}^{L} \frac{\mathrm{d}^D k_l}{i \pi^{\frac{D}{2}}} \right) \prod_{j=1}^{N} \frac{1}{P_j^{\nu_j}(\{k\},\{p\},m_j^2)}, \tag{1}$$

where the propagators $P_j$ are of the form $P_j(\{k\},\{p\},m_j^2) = q_j^2 - m_j^2 + i\delta$, with $q_j$ being a linear combination of loop momenta $k$ and external momenta $p$. Introducing Feynman parameters $x_j$ through

$$\prod_{j=1}^{N} \frac{1}{P_j^{\nu_j}} = \frac{\Gamma(\nu)}{\prod\limits_{j=1}^{N} \Gamma(\nu_j)} \int_0^{\infty} \left( \prod_{j=1}^{N} \mathrm{d}x_j \, x_j^{\nu_j-1} \right) \frac{\delta\left(1-\sum_{i=1}^{N} x_i\right)}{\left(\sum_{j=1}^{N} x_j P_j\right)^{\nu}}, \quad \text{with} \quad \nu \equiv \sum_{j=1}^{N} \nu_j, \tag{2}$$

leads to

$$G = \frac{\Gamma(\nu)}{\prod\limits_{j=1}^{N} \Gamma(\nu_j)} \int_0^{\infty} \left( \prod_{j=1}^{N} \mathrm{d}x_j \, x_j^{\nu_j-1} \right) \delta\left(1-\sum_{i=1}^{N} x_i\right) \times$$

$$\times \int_{-\infty}^{\infty} \left( \prod_{l=1}^{L} \frac{\mathrm{d}^D k_l}{i \pi^{\frac{D}{2}}} \right) \left[ \sum_{j,l=1}^{L} k_j \cdot k_l M_{jl} - 2 \sum_{j=1}^{L} k_j \cdot Q_j + J + i\delta \right]^{-\nu}. \tag{3}$$

Further details can be found e.g. in [1,37]. Integration over the momenta gives us an expression in terms of the Symanzik polynomials $\mathcal{U}$ and $\mathcal{F}$,

$$G = \frac{(-1)^{\nu} \Gamma(\nu - LD/2)}{\prod_{j=1}^{N} \Gamma(\nu_j)} \int_0^{\infty} \left( \prod_{j=1}^{N} \mathrm{d}x_j \, x_j^{\nu_j-1} \right) \delta\left(1-\sum_{l=1}^{N} x_l\right) \frac{\mathcal{U}^{\nu-(L+1)D/2}}{\mathcal{F}^{\nu-LD/2}}, \tag{4}$$

$$\text{with} \quad \mathcal{U} \equiv \det(M) \quad \text{and} \quad \mathcal{F} \equiv \det(M) \left[ \sum_{i,j=1}^{L} Q_i \left(M^{-1}\right)_{ij} Q_j - J - i\delta \right].$$

The first Symanzik polynomial, $\mathcal{U}$, is a positive semi-definite function of the Feynman parameters. The second Symanzik polynomial, $\mathcal{F}$, contains kinematic invariants and Feynman parameters. A vanishing $\mathcal{F}$ is a necessary, but not sufficient condition for infrared or kinematic singularities to arise.

**Contour deformation in pySECDEC**

After mapping all integration variables onto the unit-hypercube and integrating out the delta distribution we can absorb any additional factors by redefining and renaming $\mathcal{U} \rightarrow U$ and $\mathcal{F} \rightarrow F$. Then, eq. (4) can be written in the compact form

$$G = \int_0^1 \prod_{j=1}^{N-1} dx_j \, x_j^{\nu_j - 1} \frac{U^{\nu - (L+1)D/2}}{F^{\nu - LD/2}} = \int_0^1 \prod_{j=1}^{N-1} dx_j \, \mathcal{I}(\vec{x}) \,, \tag{5}$$

where, again, $F(\vec{x})$ can vanish inside the integration region. If we deform the integration over a Feynman parameter away from the real line segment $x \in [0,1]$ into the complex $z$-plane, Cauchy's theorem ensures that the integral does not change as long as no singularities are enclosed by the contour,

$$0 = \oint_c \prod_{j=1}^N dz_j \, \mathcal{I}(\vec{z}) = \int_0^1 \prod_{j=1}^N dx_j \, \mathcal{I}(\vec{x}) + \int_\gamma \prod_{j=1}^N dz_j \, \mathcal{I}(\vec{z})$$

$$\Leftrightarrow \int_0^1 \prod_{j=1}^N dx_j \, \mathcal{I}(\vec{x}) = -\int_\gamma \prod_{j=1}^N dz_j \, \mathcal{I}(\vec{z}) = \int_0^1 \prod_{j=1}^N dx_j \, \det\left(\frac{\partial \vec{z}(\vec{x})}{\partial \vec{x}}\right) \mathcal{I}(\vec{z}(\vec{x})). \tag{6}$$

In the complex plane the $-i\delta$ prescription in eq. (1) and eq. (4) ensures that we stay on the physical and causal Riemann sheet.

To construct an appropriate deformation into the complex plane we write $\vec{z} = \vec{x} - i\vec{\tau}$ and expand $F(\vec{z})$ around $\vec{x}$,

$$F(\vec{z}) = F(\vec{x}) - i \sum_j \tau_j \frac{\partial F(\vec{x})}{\partial x_j} - \frac{1}{2} \sum_{j,k} \tau_j \tau_k \frac{\partial^2 F(\vec{x})}{\partial x_j \partial x_k} + \frac{i}{6} \sum_{j,k,l} \tau_j \tau_k \tau_l \frac{\partial^3 F(\vec{x})}{\partial x_j \partial x_k \partial x_l} + \mathcal{O}(\tau^4). \tag{7}$$

The second term gives the leading term for the imaginary part of $F(\vec{x})$. We can guarantee that it is always negative by choosing $\tau_j \propto \partial F(\vec{x})/\partial x_j$. We also ensure that the integration endpoints are invariant under the contour deformation by requiring $\tau_j \propto x_j(1-x_j)$, defining the contour deformation as

$$\tau_j = \lambda_j x_j (1 - x_j) \frac{\partial F(\vec{x})}{\partial x_j}, \qquad \text{with} \qquad \lambda_j > 0 \,. \tag{8}$$

The *deformation parameters* $\lambda_j$ can be chosen arbitrarily, provided they are small enough for the leading order in $\tau$ to dominate the imaginary part of $F(\vec{z})$. If the $\lambda_j$ are too large the tri-linear term in eq. (7) can flip the sign of the imaginary part.

In SECDEC 3.0 [44], the $\lambda_j$ are chosen by first determining their maximal values at which the tri-linear terms in eq. (7) have the same magnitude as the linear ones; then, some fractions of these maximal values are selected via several heuristics depending on the sampled values of $\partial F(\vec{x})/\partial x_j$; a detailed description is provided in Section 6.2.3 of Ref. [48]. In pySECDEC [45] $\lambda_j$ are selected as the smallest of sampled values of

$$\left| x_j (1 - x_j) \frac{\partial F(\vec{x})}{\partial x_j} \right|^{-1} . \tag{9}$$

In both cases the initial selection is followed by iterative refinement steps: if during the integration a sign check error occurs (*i.e.* either $\operatorname{Im} F(\vec{x})$ is found to be positive or $\operatorname{Re} U(\vec{x})$ is found to be negative) for one of the sampling points, then all $\lambda_j$ are multiplied by a factor of 0.9 and the integration is repeated. As a consequence, pySECDEC often selects the largest allowed $\lambda$ vector along the initially chosen direction.

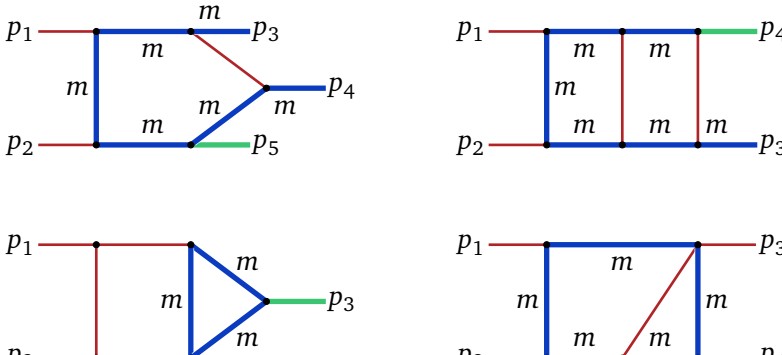

Figure 1: Feynman diagrams for our four example integrals, which we call pentagon1L, ladder2L (first line) and triangle2L, elliptic2L (second line). The blue lines denote massive lines, green lines denote massive or off-shell external legs (with a mass different from $m$).

**Example diagrams**

The Feynman diagrams we use to develop and benchmark our approaches are shown in Figure 1.

The top left diagram is a one-loop pentagon integral as it occurs in the production of a top quark pair in association with another massive particle and depends on four independent Mandelstam invariants as well as the top quark mass and the invariant mass of $p_5$. Analytically it depends on logarithms and dilogarithms of ratios of kinematic invariants, leading to a complicated branch-cut structure. After Feynman parametrization the corresponding integral is described by 4 independent Feynman parameters.

The top right diagram is a two-loop box diagram with one massive on-shell leg and one off-shell leg. This diagram is a topology occurring for example in $t\bar{t}V$ production at two loops, where the boson $V$ is radiated off an external top quark. It is close to the configuration of a 2-loop gluon ladder diagram where the exchange of gluons between two top quark lines gives rise to a Coulomb singularity. The analytic expression for this type of diagram is not known, but it is anticipated that it will contain elliptic functions. This integral depends on 6 Feynman parameters and is the most complicated example we consider in terms of dimensionality.

The diagram on the lower left of Figure 1 is a two-loop three-point function with a massive sub-triangle occurring, for instance, in NLO corrections to Higgs production in gluon fusion. It is the easiest 2-loop diagram we consider and serves as a stepping stone towards more complicated 2-loop diagrams. Analytic results for this diagram can be found in Refs. [49–51]. Depending on 5 Feynman parameters this integral is in between the previous two examples in terms of dimensionality of the integration.

The diagram on the lower right is a topology occurring in Higgs+jet production in gluon fusion at two loops. Its analytic expression contains elliptic functions and therefore is cutting edge for integrals that are currently accessible analytically. It has been calculated (semi-)analytically in Refs. [52, 53] and also served as a benchmark for the development of the program pySECDEC [45], where it is contained in the list of examples. This integral is 5-dimensional, so it has the same number of Feynman parameters as the triangle diagram, but it depends on four kinematic invariants rather than two.

## 3 Machine learning contour deformations

Numerically solving the contour integral introduced in eq. (6),

$$I = \int_0^1 \prod_{j=1}^N dx_j \, \det\left(\frac{\partial \vec{z}(\vec{x})}{\partial \vec{x}}\right) \mathcal{I}(\vec{z}(\vec{x})) \,, \tag{10}$$

with the contour deformation defined in eq. (8) still leaves the question how to choose optimal values for $\vec{\lambda}$, and the functional form is not necessarily optimal. The Monte Carlo estimate of the integral is

$$I \approx I_n = \frac{1}{n}\sum_{i=1}^n \det\left(\frac{\partial \vec{z}(\vec{x}_{(i)})}{\partial \vec{x}_{(i)}}\right) \mathcal{I}(\vec{z}(\vec{x}_{(i)})) \,. \tag{11}$$

Its statistical error is minimized if the integrand approaches a constant,

$$\det\left(\frac{\partial \vec{z}(\vec{x})}{\partial \vec{x}}\right) \mathcal{I}(\vec{z}(\vec{x})) \approx \text{const} \,. \tag{12}$$

Correspondingly, to construct an optimal contour through a neural network we use the variance of the Monte Carlo integration for large $n$ as the loss function,

$$L = \sigma_n^2 = \frac{1}{n-1}\sum_{i=1}^n \left| \det\left(\frac{\partial \vec{z}(\vec{x}_{(i)})}{\partial \vec{x}_{(i)}}\right) \mathcal{I}(\vec{z}(\vec{x}_{(i)})) - I_n \right|^2 \,. \tag{13}$$

All terms inside the absolute value squared are complex numbers. Note that the loss function has to be real valued.

### 3.1 Global complex shift

The standard pySECDEC approach of choosing the deformation parameters works fast because it only requires to evaluate $F(\vec{x})$ and its derivatives on a set of e.g. $10^4$ points, and often produces $\lambda_j$ that are good enough in practice. For challenging integrals, however, it is useful to invest time into improving the $\lambda_j$. For this purpose, we search for deformation parameters

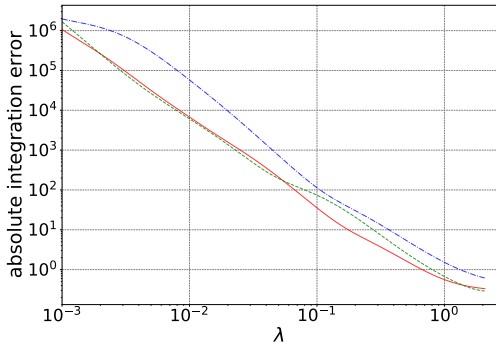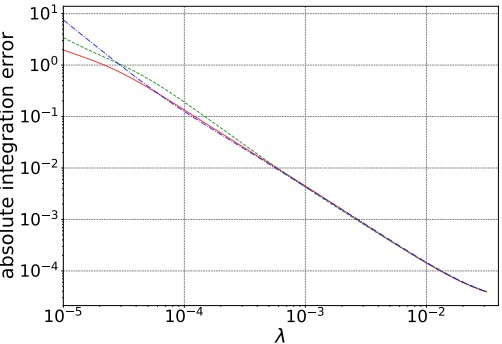

Figure 2: Absolute Monte Carlo integration errors for the first sector of the ladder2L (left) and the first sector of the elliptic2L (right) example as a function of a global $\lambda = \lambda_j$. For each case three different samples of $10^6$ $\vec{x}_{(i)}$ are shown. The plots end at the largest values of $\lambda$ that still give valid contours.

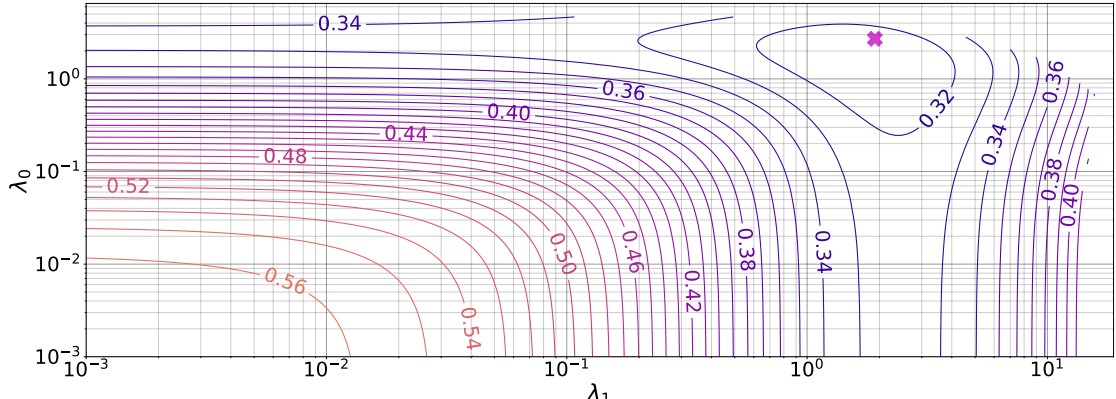

Figure 3: Absolute Monte Carlo integration error for the first sector of the ladder2L example, depending on $\lambda_0$ and $\lambda_1$, centered around $\vec{\lambda} = (2.71, 1.91, 2.73, 0.46, 5.91, 1.59)$ (marked with a cross), which is the optimum point selected by $\Lambda$-glob. Note that integration error only varies between 0.32 and 0.56 over 3 orders of magnitude in $\lambda_0$ and $\lambda_1$. We find this to be a common feature.

$\lambda_j$ which minimize the Monte Carlo integration error or the loss function $L$ on a reduced set of points.

The beneficial effect of tuning the $\lambda_j$ values is illustrated in Figure 2 and Figure 3. Two aspects complicate this minimization problem. First, the Monte Carlo integration error depends on the reference sample and can be noisy, as shown for the ladder2L example in Figure 2: different choices of the $10^6$ sampling points lead to a variation of the integration error by up to an order of magnitude. Minimizing the loss on a frozen set will overfit to the selected points and lead to an non-optimal choice for the actual integration. Second, the allowed region in the $\lambda$-space has a non-trivial shape and is currently only determined by searching for sign check errors. This noisy determination of the allowed region becomes a problem in practice, because the optimal $\lambda_j$ often lie close to this boundary, as can be seen from Figure 2. As a side remark, this is why the $\lambda$-construction in standard pySECDEC can simply choose the largest possible $\lambda_j$-vector in some predetermined direction and still work well in practice.

To circumvent these problems in optimizing the $\lambda_j$ we introduce the $\Lambda$-glob algorithm, a modified version of the Rprop algorithm [54, 55] with the added explicit handling of the allowed region for $\lambda_j$. A detailed description is given in Algorithm 1. We start with a point $\lambda_j^{(0)}$ and a step size $s_j^{(0)}$. To converge faster to a potentially far-away minimum we work with a logarithmic scale $\ell_j = \log \lambda_j$. In the first step, called *backtracking*, the algorithm decreases $\ell_j$ by some increment $\beta$ if a sign check error $\mathcal{E}_{\text{sign}}(\ell_j)$ occurs. After backtracking, we choose a proposal point $\hat{\ell}_j$ employing gradient decent. If the loss for the proposal point is smaller than for the previous point it is kept and the step size $s_j$ is decreased (increased) depending on whether the gradient of the loss has changed (not changed) its sign. If the loss is larger, the proposal point is rejected and the step size $s_j$ is decreased by a factor of $\eta^-$. All these steps are repeated a predetermined number of times. To avoid overfitting, we draw a new sample of points $\vec{x}_{(i)}$ for each iteration.

The $\Lambda$-glob algorithm benefits from a large initial step size that allows to efficiently step over local minima of $L$. Because the step size is adjusted automatically we do not have to define a learning schedule, unlike for standard gradient descent optimization.

In Figure 3 we show the landscape of the loss function and observe that our algorithm has found the global minimum and that this minimum is very broad. While it is very flat

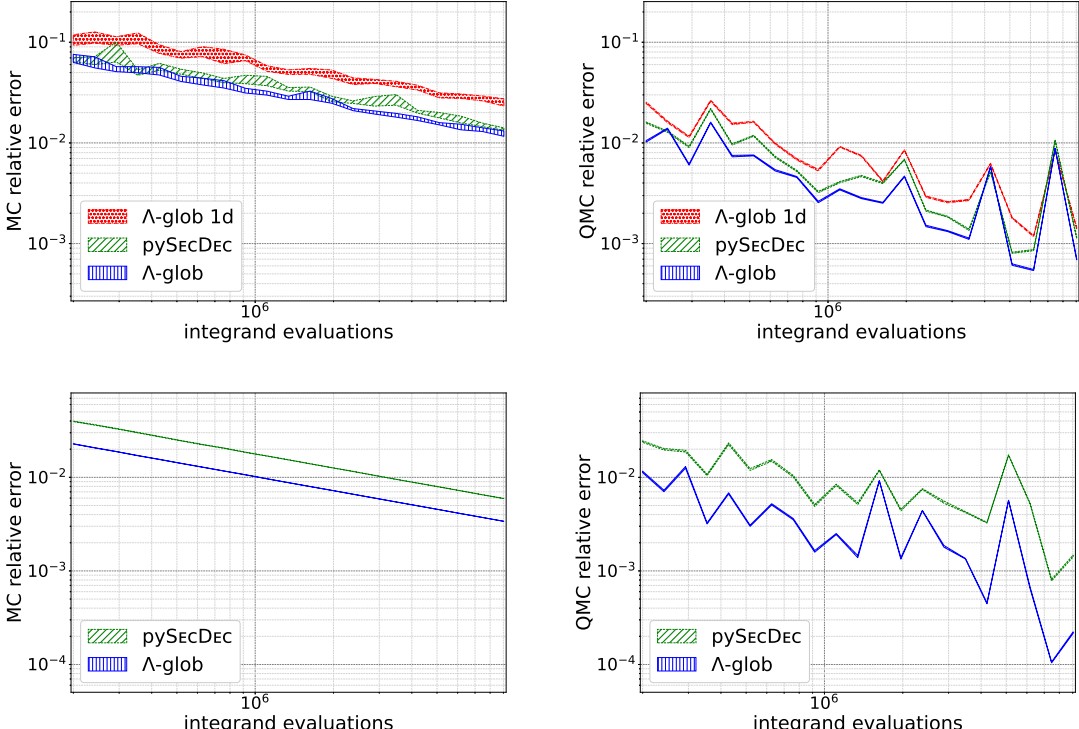

Figure 4: Monte Carlo (left) and Quasi Monte Carlo (right) integration errors for the first sector of the ladder2L example (upper) and the elliptic2L example (lower). For the ladder2L diagram we show the 1-dimensional and $n$-dimensional $\lambda$-selections using $\Lambda$-glob; for the elliptic2L diagram these two turn out to be the same, so only one is shown.

in $\lambda_0$ and $\lambda_1$ individually, a correlated shift increases the loss function more steeply. Results of this algorithm for different Feynman integrals are presented in Figure 4. In addition to a standard Monte Carlo algorithm we also show the result for the Quasi Monte Carlo algorithm in pySecDec [46]. Depending on the integral and the kinematic configuration $\Lambda$-glob gives comparable or improved results compared to the standard pySecDec construction.

## 3.2 Generalized local transformation

Moving beyond the optimization of a global deformation parameter, we can exploit the full freedom of the reparametrization with a local transformation of $\lambda$ to further minimize the Monte Carlo error. In principle any reparametrization in eq. (8) would serve our purpose as long as the defined contour does not cross or enclose any singularities. However, a good contour should also obey the following criteria:

1. The Monte Carlo error should be minimized, *i.e.* the product of the Jacobian of the transformation and the integrand should be nearly constant, see eq. (12);

2. The endpoints have to be fixed for Cauchy's theorem to be applicable as in eq. (6);

3. The parametrization should be numerically stable and, if possible, have tractable Jacobians. A tractable Jacobian is not only more stable numerically, it also helps to make sure that the procedure gives meaningful results.

Let us consider again the integral along a contour $\gamma$,

$$I = \int_\gamma \prod_{j=1}^{N} \mathrm{d}z_j \, \mathcal{I}(\vec{z}) \,. \tag{14}$$

We parametrize this contour in terms of the real parameters $y_j \in [0,1]$,

$$z_j = y_j - \mathrm{i}\tau_j(\vec{y}) \qquad \text{and} \qquad \tau_j = \lambda_j(\vec{y}) \, y_j (1 - y_j) \frac{\partial F(\vec{y})}{\partial y_j} \,, \tag{15}$$

where the form of the imaginary part guarantees the correct boundary conditions. In contrast to eq. (8), the deformation $\lambda_j$ is now a *local* parameter, depending on $\vec{y}$.

To minimize the variance in the numerical integration, $\vec{y}$ needs to be sampled according to some non-trivial probability distribution. In practice, this probability distribution is neither known nor is it possible to easily sample from it. Therefore, we introduce an additional mapping

$$y_j = f_j(\vec{x}) \,, \tag{16}$$

---

**Algorithm 1** The $\Lambda$-glob algorithm for the global lambda optimization. Good default settings for the tested examples were $\lambda_j^{(0)} = 0.1$, $s_j^{(0)} = 2$, $\eta^- = 0.5$, $\eta^+ = 1.125$, $\beta = 0.1$, and $n = 100$.

---

**Require:** $L(\vec{x}, \ell_j)$: The loss function (given by e.g. eq. (13)) with parameters $\lambda_j = e^{\ell_j}$
**Require:** $\mathcal{E}_{\mathrm{sign}}(\vec{x}, \ell_j)$: Sign check error boolean function with parameters $\lambda_j = e^{\ell_j}$
**Require:** $s_j^{(0)}$: The initial step size vector
**Require:** $0 < \eta^- < 1 < \eta^+$: Step size decrease and increase factors
**Require:** $\beta > 0$: The size of the backtracking step
**Require:** $\lambda_j^{(0)}$: Initial parameter vector
**Require:** $n$: The number of the optimization iterations
  $\ell_j^{(0)} \leftarrow \log \lambda_j^{(0)}$ (Initialize log parameter)
  **for** $t = 1, \ldots, n$ **do**
    Draw a sample $\vec{x}^{(t)}$ from a unit hypercube
    $\widehat{\ell}_j \leftarrow \ell_j^{(t-1)}$
    **while** $\mathcal{E}_{\mathrm{sign}}(\vec{x}^{(t)}, \widehat{\ell}_j)$ **do**
      $\widehat{\ell}_j \leftarrow \widehat{\ell}_j - \beta$
    **end while**
    $\widehat{\ell}_j \leftarrow \widehat{\ell}_j - s_j^{(t-1)} \mathrm{sgn}\!\left( \frac{\partial L}{\partial \ell_j}\!\left(\vec{x}^{(t)}, \widehat{\ell}_j\right) \right)$
    **if** $L\!\left(\vec{x}^{(t)}, \widehat{\ell}_j\right) > L\!\left(\vec{x}^{(t)}, \ell_j^{(t-1)}\right)$ **then**
      $\ell_j^{(t)} \leftarrow \ell_j^{(t-1)}$
      $s_j^{(t)} \leftarrow \eta^- s_j^{(t-1)}$
    **else**
      $\ell_j^{(t)} \leftarrow \widehat{\ell}_j$
      $s_j^{(t)} \leftarrow \left( \text{if } \left(\frac{\partial L}{\partial \widehat{\ell}_j}\right) \cdot \left(\frac{\partial L}{\partial \ell_j^{(t-1)}}\right) > 0 \text{ then } \eta^+ \text{ else } \eta^- \right) s_j^{(t-1)}$
    **end if**
  **end for**
  $\lambda_j \leftarrow \exp\!\left(\ell_j^{(n)}\right)$
  **return** $\lambda_j$

---

with uniformly distributed $x_j \in [0, 1]$. With all these transformations the integral becomes

$$
\int_\gamma \prod_{j=1}^N \mathrm{d}z_j \, \mathcal{I}(\vec{z}) = \int_0^1 \prod_{j=1}^N \mathrm{d}y_j \, \det\left(\frac{\partial \vec{z}(\vec{y})}{\partial \vec{y}}\right) \mathcal{I}(\vec{z}(\vec{y}))
$$
$$
= \int_0^1 \prod_{j=1}^N \mathrm{d}x_j \, \det\left(\frac{\partial \vec{z}(\vec{y})}{\partial \vec{y}}\right) \det\left(\frac{\partial \vec{y}(\vec{x})}{\partial \vec{x}}\right) \mathcal{I}(\vec{z}(\vec{y}(\vec{x}))) \,.
$$
(17)

Except for the boundaries, the functions $\lambda$ and $f$ can be chosen freely. A flexible and promising way to parametrize these functions is with neural networks. A critical aspect of the reparametrization are the Jacobians

$$
(J_\lambda)_{jk} = \frac{\partial z_j}{\partial y_k} = \frac{\partial (y_j - \mathrm{i}\tau_j)}{\partial y_k} = \delta_{jk} - \mathrm{i}\frac{\partial \tau_j}{\partial y_k} \qquad \text{and} \qquad \left(J_f\right)_{jk} = \frac{\partial y_j}{\partial x_k} = \frac{\partial f_j(\vec{x})}{\partial x_k} \,,
$$
(18)

the first of which is complex. For these Jacobians to be non-singular, we require our mappings to be bijective. While the complex Jacobian is always non-singular by construction, we have to ensure explicitly that the function $f$ is bijective. The function $\lambda$ does not have to be bijective. However, one needs to ensure that the sub-Jacobian

$$
\frac{\partial \tau_j}{\partial y_k} = \frac{\partial \lambda_j(\vec{y})}{\partial y_k} y_j(1 - y_j) \frac{\partial F(\vec{y})}{\partial y_j} + \lambda_j(\vec{y}) \delta_{jk} (1 - 2y_j) \frac{\partial F(\vec{y})}{\partial y_j} + \lambda_j(\vec{y}) y_j(1 - y_j) \frac{\partial^2 F(\vec{y})}{\partial y_j \partial y_k} \,,
$$
(19)

is numerically stable.

### 3.3 Normalizing flow setup

The reasons to split the full mapping $\vec{x} \to \vec{z}$ into the real mapping $\vec{x} \to \vec{y}$ and the complex mapping $\vec{y} \to \vec{z}$ as in eq. (15) and eq. (16) are the following: First, it will allow us to use a normalizing flow [56–59] for the real mapping, giving us a tractable Jacobian. The Jacobian of the complex mapping needs to be evaluated numerically and is computationally more expensive than the Jacobian of the normalizing flow. Second, when we evaluate a kinematic phase-space point which does not require any contour deformation, as there are no integrable singularities, we just turn off the complex mapping. The real mapping then becomes a version of neural importance sampling [4–9].

To train our network we will use the variance loss defined in eq. (13). From the discussion of the $\Lambda$-glob algorithm and Figure 2 we know that the integration error is fairly insensitive to small changes in $\lambda$. Furthermore, for the parametrization in eq. (15), we found that making $\lambda$ local (i.e. dependent on $\vec{x}$) also hardly affects the loss. Because the introduction of a neural network comes with a computational cost, we keep $\lambda$ global in our NN-approach. This means we rely on the $\Lambda$-glob algorithm to first find optimized $\lambda_j$ and then use a normalizing-flow network to optimize the sampling of the real parameters and minimize the variance. In our experiments we perform the numerical loop integration for various Feynman diagrams given in Figure 1, which are represented in the $N$-dimensional Feynman parameter space.

### Network architecture

Normalizing flows encode a bijective mapping between a physics and a latent space. The model can be evaluated in either direction with comparable efficiencies, at least in the invertible network (INN) variant [60–62]. Even if we are not interested in this symmetric evaluation,

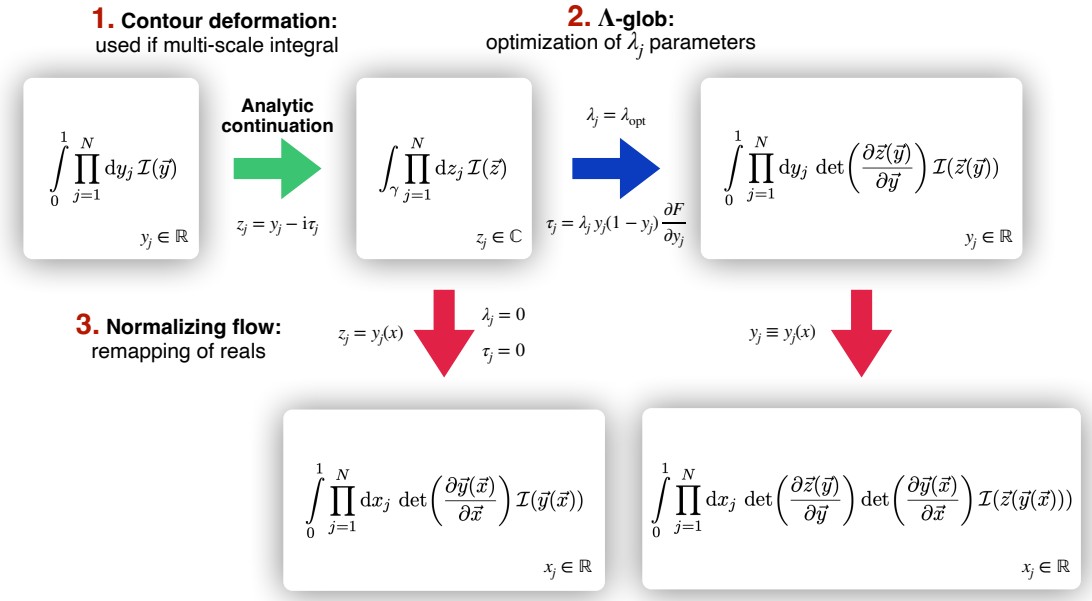

Figure 5: Schematic illustration of our workflow.

normalizing flows have the considerable advantage of a tractable Jacobian. A simple realization are stacked coupling layers [61, 63], where we split the input vector $x$ in $x_1$ and $x_2$ and use an element-wise multiplication $\odot$ and sum to define the mapping

$$
\begin{aligned}
y_1 &= x_1 \odot e^{s_1(x_2)} + t_1(x_2), & x_1 &= (y_1 - t_1(x_2)) \odot e^{-s_1(x_2)}, \\
y_2 &= x_2 \odot e^{s_2(y_1)} + t_2(y_1), & x_2 &= (y_2 - t_2(y_1)) \odot e^{-s_2(y_1)},
\end{aligned}
\tag{20}
$$

where $s_1, s_2, t_1$ and $t_2$ are parametrized by neural networks. The Jacobian of such a coupling block is [61]

$$
J = \begin{pmatrix} \mathbb{1} & 0 \\ \frac{\partial y_2}{\partial y_1} & \mathrm{diag}(e^{s_2(y_1)}) \end{pmatrix} \begin{pmatrix} \mathrm{diag}(e^{s_1(x_2)}) & \frac{\partial y_1}{\partial x_2} \\ 0 & \mathbb{1} \end{pmatrix}.
\tag{21}
$$

While $J$ is not triangular, we will only be interested in the log-determinant, which can be calculated efficiently as

$$
\log(\det J) = \log\left( \prod_{i=1}^{\dim x_2} e^{s_1(x_2)_i} \right) + \log\left( \prod_{i=1}^{\dim y_1} e^{s_2(y_1)_i} \right) = \sum_{i=1}^{\dim x_2} s_1(x_2)_i + \sum_{i=1}^{\dim y_1} s_2(y_1)_i.
\tag{22}
$$

For all examples we employ a normalizing flow consisting of these affine coupling blocks, where each coupling block describes a bijective mapping $\mathbb{R}^N \leftrightarrow \mathbb{R}^N$. To map the Feynman parameters $x \in [0,1]^N$ from the unit-hypercube to $\mathbb{R}^N$ bijectively we apply the logit function

$$
y = \mathrm{logit}(x) \equiv \log\left( \frac{x}{1-x} \right), \qquad \text{with} \qquad \left( J_{\mathrm{logit}} \right)_{jk} = \frac{\delta_{jk}}{x_j - x_j^2},
\tag{23}
$$

which is the inverse of the sigmoid function

$$
y = \mathrm{sig}(x) \equiv \frac{1}{1 + \exp(-x)}, \qquad \text{with} \qquad \left( J_{\mathrm{sig}} \right)_{jk} = \delta_{jk}\, \mathrm{sig}(x_j)(1 - \mathrm{sig}(x_j)).
\tag{24}
$$

As both Jacobians are diagonal these functions can be easily combined with the coupling blocks. For convenience, we use the sigmoid as the final network layer, such that the output domain is again the unit-hypercube. We sandwich 14 coupling blocks between the logit and sigmoid functions. In each coupling block we use a simple fully connected neural network consisting of 3 layers with 128 units and Leaky ReLU as activation function. To regularize the exponentials of the affine coupling block we use soft clamping [63], $s_{\text{clamp}} = c \cdot \tanh(s)$, with $c = 0.5$, and activation normalization [62]. Furthermore, we use random orthogonal matrices [64] to allow for more interaction between the two parts $y_1, y_2$ in the coupling blocks. Our network is implemented using TENSORFLOW [65].

**Training**

Before introducing the neural network, we employ the $\Lambda$-glob algorithm to find optimal values of $\lambda_j$, which minimize the variance loss in eq. (13) and define a valid contour on the physical Riemann sheet. Next, we train the normalizing flow to re-sample the real parameters in the spirit of neural importance sampling. This gives us a complex mapping $\vec{y} \rightarrow \vec{z}$ parametrized as in eq. (8) with optimized $\lambda_j$, and a real mapping $\vec{x} \rightarrow \vec{y} = f(\vec{x})$ where $f$ is represented by a normalizing flow.

For kinematic phase-space regions below threshold, no contour deformation is needed. Here the $\Lambda$-glob algorithm will find $\lambda_j = 0$, the complex mapping eq. (8) will be omitted, and the real mapping alone will improve the calculation. The complete workflow is summarized in Figure 5.

In contrast, for kinematic phase-space points above threshold the contour is vital and we need to make sure to have the correct sign for the imaginary part of $F$ as well as for the real part of $U$. For the $\Lambda$-glob algorithm we use a simple backtracking method to discard a proposal state and step back, *i.e.* reduce the value of $\lambda_j$, if a sign check-error occurs. For the network we add a term to the loss function. As we employ the ADAM optimizer [66], this sign loss has to be differentiable, so we add

$$L_{\text{sign}} = Y \operatorname{sig}\left(\frac{\operatorname{Im} F}{X_F}\right) + \operatorname{ReLU}(\operatorname{Im} F) + Y \operatorname{sig}\left(-\frac{\operatorname{Re} U}{X_U}\right) + \operatorname{ReLU}(-\operatorname{Re} U), \qquad (25)$$

with $\operatorname{ReLU}(x) = \max\{0, x\}$ to the variance loss of eq. (13). Using a validation set $\vec{x}_{\text{val}}$, the relative scales $Y, X_F, X_U$ are estimated in the beginning of the training and updated every $K^{\text{th}}$

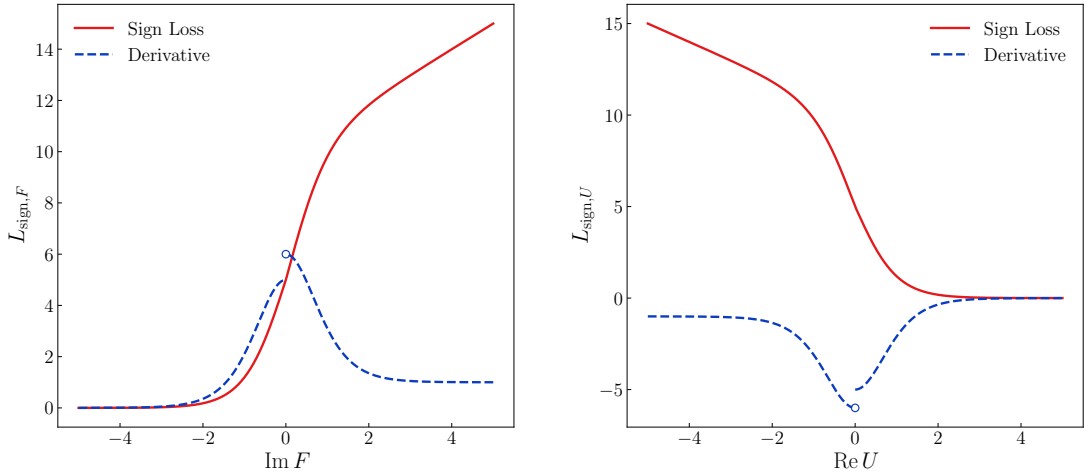

Figure 6: Illustration of the sign loss and its derivative for the $F$ (left) and $U$ (right) part for $Y = 10$ and $X_{\{F,U\}} = 1/2$.

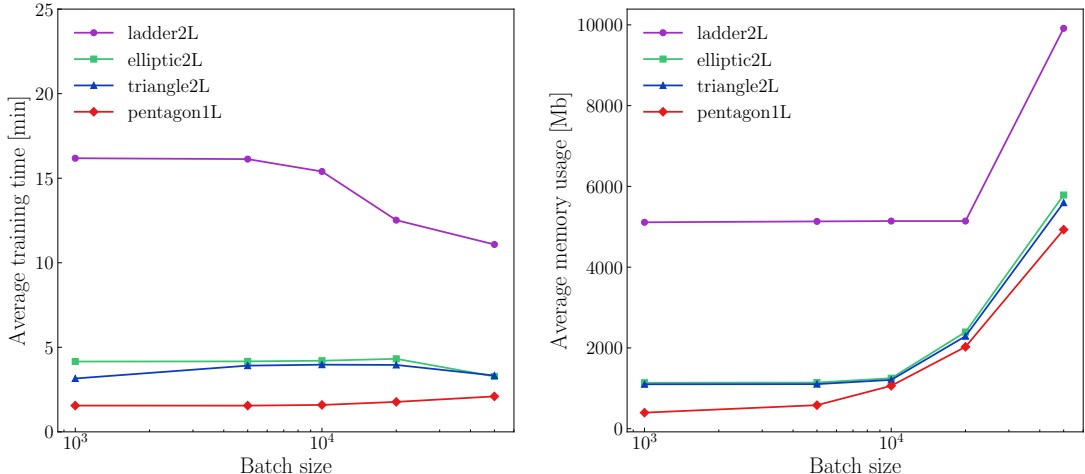

Figure 7: Average training times (left) and average memory consumption (right) for a single phase-space point as a function of the batch size $b$.

iteration according to

$$Y = 10 \cdot L_{\text{var}}(\vec{x}_{\text{val}}), \quad X_F = \frac{1}{5n}\left|\sum_i^n \text{Im}\, F(\vec{x}_{\text{val},(i)})\right|, \quad X_U = \frac{1}{5n}\left|\sum_i^n \text{Re}\, U(\vec{x}_{\text{val},(i)})\right|. \quad (26)$$

In practice, we find that $K = 10$ works well in our experiments. An illustration of the sign loss and its derivative of the $F$ and $U$ part is shown in the left and right panels of Figure 6, respectively.

Moreover, a numerical bottleneck in our contour optimization is the calculation of the complex-valued determinant and its derivative. As the TENSORFLOW implementation of complex-valued determinants yields wrong gradients[1], we implement our own version of the determinant. It relies on the recursive Laplace expansion and becomes computationally expensive for higher dimensional cases. This can be seen in the GPU-memory usage in Figure 7, which is significantly higher for processes involving more Feynman parameters, such as the ladder2L example. This is one of the reasons why the timings are not competitive with the timings for the standard contour deformation in pySECDEC. The largest benefit from the ML-approach is expected for high-dimensional multi-scale cases, where the contour avoiding all poles and branch cuts is a highly non-trivial hypersurface in the complex integration space. In such cases the gain in numerical precision can be so large that it outweighs the time spent to train the network. Indeed, the true advantage would show up in calculations of complete amplitudes, rather than individual integrals, containing a few integrals that would barely converge at all in pySECDEC but would converge well with an optimized contour.

**Performance**

Finally, we illustrate the performance gain achieved by applying both, the $\Lambda$-glob algorithm only and its combination with the normalizing flow.

In Figure 8 we show results for the triangle2L (left) and the elliptic2L (right) integral. For both integrals we consider the first sector integral after sector decomposition. We sample 100 phase space points varying over 4-5 orders of magnitude in the squared center-of-mass energy $s \equiv (p_1 + p_2)^2$. For both processes, we intentionally consider points below and above threshold,

---

[1]See the issue raised at GITHUB: https://github.com/tensorflow/tensorflow/issues/49946.

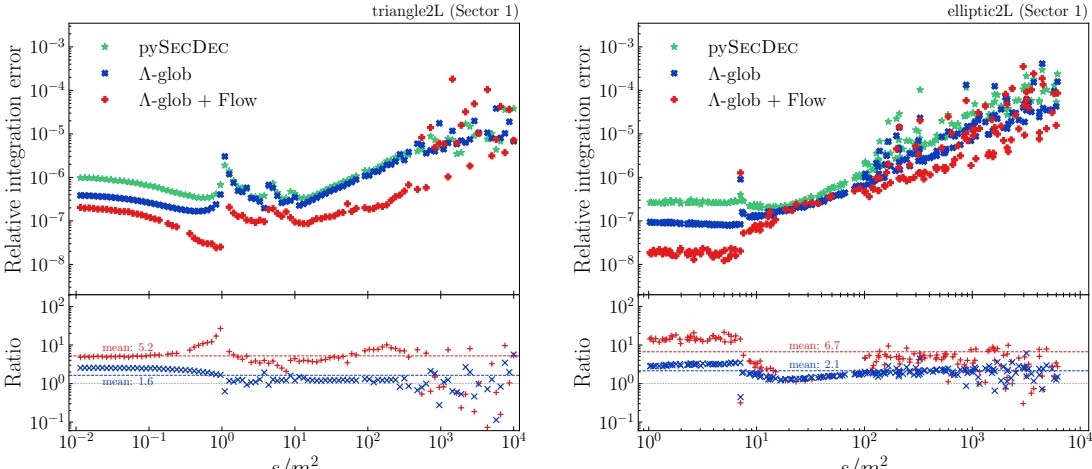

Figure 8: Relative integration error for sector one of the triangle2L (left) and elliptic2L (right) integrals using the standard pySecDec algorithm (green), the $\Lambda$-glob algorithm (blue) and $\Lambda$-glob with additional normalizing flow (red). The lower panel shows the ratios to the standard method.

to compare the performance when no contour deformation is needed. We normalized the kinematic invariants using $m^2 = 1$. For the triangle2L integral, shown in the left panel of Figure 8, the average integration error over all phase-space points is reduced by a factor two for the $\Lambda$-glob algorithm and by a factor of 5 for our ML-approach. In the low-energy regime the error reduction stays around the average value. For increasing energies towards threshold at $s/m^2 = 1$, the absolute integration error of the standard pySecDec method and the pure $\Lambda$-glob algorithm increase, while absolute integration error of our ML-approach keeps decreasing. This results in a relative performance gain by a factor of up to 30 close to the threshold. The threshold being located at $s/m^2 = 1$ is a consequence of considering sector one, which effectively corresponds to a topology where one of the massive triangle propagators connecting to $p_3$ is pinched. In contrast, in the elliptic2L sector 1 integral, shown in the right panel of Figure 8, the importance sampling through the normalizing flow reduces the integration error by a factor of 20 and does not show the rising profile towards the threshold. The average integration error is reduced by a factor of 7 or 2 depending on whether the additional mapping of the normalizing flow is used or not. The kinematic points for this diagram are chosen to have varying values of $t = (p_1 + p_3)^2$ and $p_4^2$.

In general, for energies close but above threshold the performance gain is less pronounced, as the contour deformation in this regime has less freedom for optimization and the effect of modifying the real parts is diminished.

For increasing energies, the absolute integration error also increases and eventually starts fluctuating. This is driven by the singularities moving toward the endpoints. A possible way to control this behavior has been proposed in Ref. [47]. Together with the absolute integration error, the improvement factors also start to fluctuate strongly for large energies.

Finally, in Figure 9 we show the results for the more complicated pentagon1L (left) integral and the ladder2L (right) integral. Again, for both integrals we consider the first sector integral after sector decomposition. The increasing complexity originates from both a higher-dimensional integration space, *i.e.* more Feynman parameters, and from having more kinematic scales involved. In order to cover possible dependencies on other kinematic variables than $s$ and $m^2$, we decided to sample different kinematic phase-space for the same values of $s/m^2$. For both integrals we find that the average integration error reduces by a factor two for

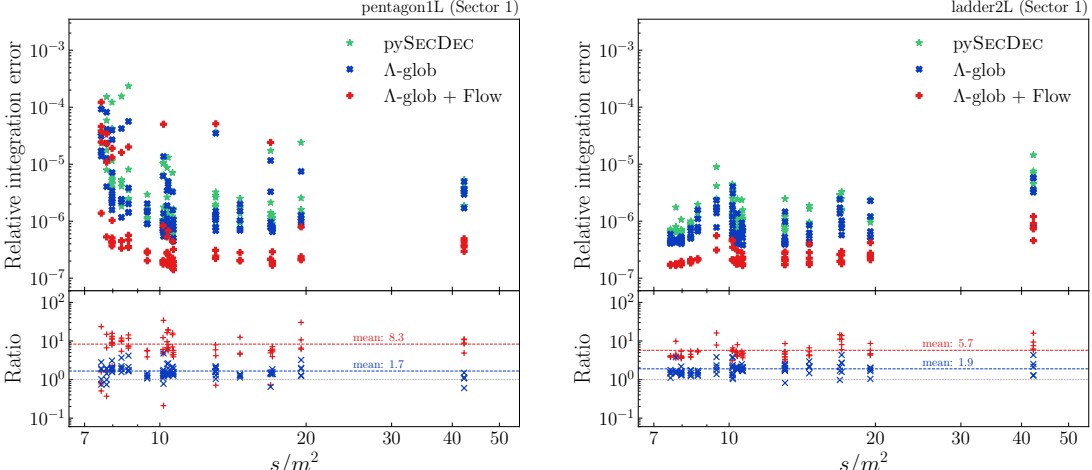

Figure 9: Relative integration error for sector one of the pentagon1L (left) and ladder2L (right) integrals using the standard pySECDEC algorithm (green), the $\Lambda$-glob algorithm (blue) and $\Lambda$-glob with additional normalizing flow (red). The lower panel shows the ratios to the standard method.

the $\Lambda$-glob algorithm. By employing the ML-method we achieve an average error reduction factor of 6 and 8 for the ladder2L and pentagon1L, respectively. For individual phase-space points we achieve an improvement factor of up to 30. However, there are also phase-space points for which both the $\Lambda$-glob and the flow supplemented algorithm show inferior performance. This clearly indicates the shortcomings of the optimization procedures which are related to the strict sign requirement on the imaginary part.

## 4 Outlook

We have shown, for the first time, that the application of modern machine learning methods to numerical multi-loop calculations can lead to a considerable reduction of the numerical uncertainties and hence speed. This has been achieved in a two-step procedure, first applying an algorithm to globally optimize the contour deformation parameters $\lambda$, and subsequently employing a normalizing flow to optimize the complex integration contour, after splitting the full contour deformation into a real and an imaginary part. We have demonstrated the performance with several one- and two-loop examples. All of these examples contain massive propagators and several kinematic scales, leading to a complicated threshold structure of the integrand, such that the contour deformation is a highly non-trivial task, which was dealt with successfully by the neural networks. While the results presented in this paper can only be a first step, they very much motivate further investigations.

## Acknowledgements

We would like to thank Margarete Mühlleitner for useful discussions. RW acknowledges support by HeiKA and by FRS-FNRS (Belgian National Scientific Research Fund) IISN projects 4.4503.16. The research of AB, GH and TP is supported by the Deutsche Forschungsgemeinschaft (DFG, German Research Foundation) under grant 396021762 — TRR 257 *Particle Physics Phenomenology after the Higgs Discovery*. This work was supported by the Deutsche

Forschungsgemeinschaft (DFG, German Research Foundation) under Germany's Excellence Strategy EXC 2181/1 - 390900948 (the Heidelberg STRUCTURES Excellence Cluster).

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
