# Peer review of "Targeting Multi-Loop Integrals with Neural Networks"

_SciPost Physics, doi:SciPost Phys. 12, 129 (2022)_

## Round 1 · Referee Report · Anonymous (Referee 1) · 2022-2-24

Report

The authors have sufficiently addressed all of the concerns about their work. The work easily meets the criteria for SciPost. I recommend that the article be published.

---

## Round 1 · Author Response

We would like to thank the referee for the constructive report. We have produced a new version of the paper that integrates the referee’s suggestions. Below is a detailed report, with our responses offset by “>>>”. Sincerely, the authors

Requested changes

1 - In the Example diagrams section, the authors describe an example as to where the diagram would show up in calculations. However, the authors do not do so for the lower left diagram of Figure 1. The authors should add this description. >>> We added a sentence about a process relevant at the LHC where this diagram shows up in calculations. 2 - In the first paragraph of Section 3.3, the authors mention that the Jacobian of the complex mapping is computationally more expensive. However, it is unclear to what the computation is more expensive than. Is it the normalizing flow? The original method? >>> It is more expensive compared to the tractable Jacobian of the normalizing flow. We clarified this in the text now. 3 - Figure 7 is referenced before Figure 6. The authors should switch the ordering to match the order they are introduced. >>> We changed the order of Figures 6 and 7 in the new version. 4 - The authors make a comment about the timings not being competitive with pySecDec, with the complex-valued determinants being one of the driving factors. And again mention the improvement in speed in the outlook. However, the authors make no comparison of the timings. It would be informative to the reader to have a plot showing the time required to achieve a given precision. This would be helpful in understanding when this algorithm might be better than default pySecDec. >>> At this level, we decided not to make a quantitative comparison because, for the examples considered which were simple enough to serve for development purposes, the default pySecDec algorithm would always win in a strict timing comparison. Furthermore, it is unclear at what level the timings should be compared, e.g. comparing only the integration times or including the training times. However, as we pointed out already in the text, we expect the NN-approach to outperform the default algorithm for high-dimensional multi-scale cases in particular kinematic regions, where the valid hypersurfaces are very difficult to optimize. Such cases would show up in the calculation of full amplitudes rather than individual integrals. Therefore, in the new version, we added a sentence that the true advantage of the proposed method would arise in the calculation of complicated multi-loop amplitudes, where the default algorithm in some kinematic regions may not converge at all.

---

## Round 1 · List of Changes

see above

---

## Editorial Decision

published